# Emerging Role of AP-1 Transcription Factor JunB in Angiogenesis and Vascular Development

**DOI:** 10.3390/ijms22062804

**Published:** 2021-03-10

**Authors:** Yasuo Yoshitomi, Takayuki Ikeda, Hidehito Saito-Takatsuji, Hideto Yonekura

**Affiliations:** Department of Biochemistry, Kanazawa Medical University School of Medicine, 1-1 Daigaku, Uchinada, Kahoku-gun, Ishikawa 920-0293, Japan; tikeda@kanazawa-med.ac.jp (T.I.); saitoh@kanazawa-med.ac.jp (H.S.-T.); yonekura@kanazawa-med.ac.jp (H.Y.)

**Keywords:** AP-1 transcription factors, JunB, angiogenesis, tip cell specification, vascular development, neurovascular interactions

## Abstract

Blood vessels are essential for the formation and maintenance of almost all functional tissues. They play fundamental roles in the supply of oxygen and nutrition, as well as development and morphogenesis. Vascular endothelial cells are the main factor in blood vessel formation. Recently, research findings showed heterogeneity in vascular endothelial cells in different tissue/organs. Endothelial cells alter their gene expressions depending on their cell fate or angiogenic states of vascular development in normal and pathological processes. Studies on gene regulation in endothelial cells demonstrated that the activator protein 1 (AP-1) transcription factors are implicated in angiogenesis and vascular development. In particular, it has been revealed that JunB (a member of the AP-1 transcription factor family) is transiently induced in endothelial cells at the angiogenic frontier and controls them on tip cells specification during vascular development. Moreover, JunB plays a role in tissue-specific vascular maturation processes during neurovascular interaction in mouse embryonic skin and retina vasculatures. Thus, JunB appears to be a new angiogenic factor that induces endothelial cell migration and sprouting particularly in neurovascular interaction during vascular development. In this review, we discuss the recently identified role of JunB in endothelial cells and blood vessel formation.

## 1. Vascular Endothelial Cells and Activator Protein 1 (AP-1) Transcription Factors

### 1.1. Endothelial Cell Heterogeneities and Gene Expression

Vascular endothelial cells represent the principal cells of blood vessels in most tissues. They display heterogeneity and different characteristics depending on the state of angiogenesis and tissue type. The differences noted between vascular endothelial cells include the basic properties of arteries, veins, capillaries, tip cells, and stalk cells. In addition, they include a wide variety of tissue-specific endothelial cells, such as the blood-brain barrier structure bearing cerebral blood vessels, and liver sinusoidal vascular endothelial cells that have a loose basement membrane structure. It has been shown that these differences are related to differences in gene expression. For example, blood-brain barrier transporters major facilitator superfamily domain containing 2A (Mfsd2a) and solute carrier family 2 member 1 (Slc2a1) are specifically expressed in endothelial cells of the central nervous system [1]. Moreover, GATA binding protein 4 (GATA4) is involved in the formation of sinusoidal blood vessels in the liver [2]. There is accumulating evidence regarding gene expression in vascular endothelial cells. Recently, a large-scale transcriptome analysis of tissue-type vascular endothelial cells isolated from various tissues identified various tissue-specific vascular endothelial cell gene transcriptomes [3,4,5]. These data are available in the public vascular endothelial cell transcriptome database EndDB, hosted by VIB-KU Leuven Center for Cancer Biology (Leuven, Belgium; URL: https://vibcancer.be/software-tools/endodb, accessed on 1 February 2021) [6]. The transcription of vascular endothelial cells, similar to that of other cells, is regulated by a number of transcription factors and epigenetic regulation. A recent comprehensive analysis of the chromatin states of human vascular endothelial cells identified 3765 endothelial-specific enhancers [7]. They also identified nine endothelial cell groups divided into two subgroups based on the epigenomic landscape. In addition, numerous homeobox genes and some other transcription factors were differentially activated across the endothelial cell types [7].

The process of vascular development, termed vasculogenesis, is initiated at the early developmental stages by the formation of a primitive vascular plexus, a beehive-like structure of endothelial progenitor cells. In the process of angiogenesis, the primordial vascular plexus invades the vascular-free area in response to angiogenic cues, such as hypoxia and vascular-inducing factors. Budding, branching, and fusion occur repeatedly to create more complex capillary networks. The vascular endothelial growth factor (VEGF) is a primary regulator of angiogenesis and blood vessel formation that controls endothelial cell proliferation, survival, and migration to form blood vessels. VEGFA controls angiogenic sprouting by guiding filopodia extension from tip cells at the vascular-sprouting frontier [8]. At the protrusion tip of the vascular elongation, the first cells which receive VEGF signals, become “tip cells” through VEGF intracellular signaling, thereby forming numerous filopodia and enhancing cell motility. In addition, tip cells express the Notch ligand delta-like canonical Notch ligand 4 (DLL4). Moreover, vascular endothelial cells adjacent to tip cells bind to their membrane receptor Notch1, which transmits signals into the cells and causes them to become stalk cells. Stalk cells lack filopodia, are proliferative, and regulate the number of cells in the subsequent vascular network [8]. Tip and stalk cells maintain plasticity in the formation of the vascular network. Stalk cells may become tip cells or, conversely, tip cells may degenerate into stalk cells by retracting their filopodia. Thus, DLL4-Notch1 signaling regulates the specification of tip cells and stalk cells to maintain proper vessel density [9]. From the primordial vascular plexus, through a process termed remodeling, mature vessels with hierarchical structure are finally constructed in each tissue and organ. These steps include arterial and venous specification, vessel remodeling in the coordination with neurovascular parallel alignment, and formation of functional blood vessels. Numerous previous studies have described the regulation of VEGFA gene expression. Many transcription factors have been identified as VEGF-positive and -negative regulators, including hypoxia-inducible factor 1 subunit alpha (HIF1a) [10,11], Sp transcription factors [12], NF-κB [13], SMAD [14], SRY-box transcription factor 9 (SOX9) [15], forkhead box O3 (FOXO3) [16,17], signal transducer and activator of transcription 3 (STAT3) [18,19], cAMP responsive element binding protein 1 (CREB1) [20], and AP-1 transcription factors [21,22,23,24,25,26,27,28]. In recent years, research has focused on the function of AP-1 factors in endothelial cells. In this review, we focused on the AP-1 transcription factor JunB in endothelial cells in VEGF signaling during angiogenesis.

### 1.2. AP-1 Transcription Factors in Endothelial Cells

The AP-1transcription factor family consists of Jun (c-Jun, JunB, and JunD), Fos (c-Fos, FosB, Fra-1, and -2), and ATF (ATF-2, -3, -4, and ATFa). These factors form homodimers or heterodimers at the N-terminal leucine zipper common motif, which bind to DNA through the DNA binding motif, thereby regulating the transcription of target genes [29,30,31]. AP-1 factors control both the basal and inducible transcription of several genes which contain AP-1 sites (consensus sequence 5′-TGAG/CTCA-3′) on their promoter. These consensus sequences are also termed 12-O-tetradecanoylphorbol-13-acetate-responsive elements [29]. In the case of Jun, it is known that the heterodimer Jun/Fos has higher DNA affinity and transcriptional activity than the homodimer Jun/Jun, suggesting that Jun functions as a heterodimer in vivo [30,32]. AP-1 transcription factors are involved in numerous physiological and pathological processes, including the cell cycle, development, and tumor progression. AP-1 transcription factors are also known as immediately early genes, which are transiently and rapidly activated in response to a wide variety of cellular stimuli. These genes are involved in the regulation of gene activity following the primary growth factor response, including VEGFs [33,34]. AP-1 transcription factors are reportedly involved in regulating VEGF expression and endothelial cell gene expression in response to VEGF stimulation [35,36]. In endothelial cells, VEGF stimulation induces its target genes during angiogenesis, and AP-1 transcription factors regulate the expression of these genes by binding to their promoters [26,37,38,39]. AP-1 transcription factors also reportedly regulate gene expressions implicated in angiogenesis, including VEGFs and matrix metalloproteinases (MMPs) [40,41,42,43].

## 2. AP-1 Transcription Factor JunB in Angiogenesis

### 2.1. JunB Expression in Endothelial Cells

Previously, it was found that JunB is involved in the differentiation of erythroid cells [44] and T cells [45], as well as tadpole tail regeneration by positively regulating cell proliferation [46]. JunB also positively regulates the proliferation of embryonic fibroblast cells by promoting S to G2/M transition through cyclin A activation [47]. However, the negative regulation of cell proliferation by inhibition of G1-S transition in HeLa cells through inhibition of the cyclin D1 promoter has also been reported [48]. In human umbilical vein endothelial cells (HUVEC), dominant negative c-Fos blocks endothelial cell proliferation by inhibiting cyclin D expression and also inhibits cell migration. In contrast, JunB knockdown in HUVEC attenuated endothelial cell migration but did not affect the proliferation of endothelial cells [33]. These findings indicated that JunB is required primarily for cell migration but may not control the proliferation of endothelial cells.

Expression of JunB in human tissues/organs during different developmental, normal, and pathological conditions has been described. Moreover, human placental JunB expression in JunB/cyclin-D1 imbalance in placental mesenchymal stromal cells derived from pre-eclamptic pregnancies with fetal placental complications has been described [49]. Amplification and overexpression of JunB are associated with primary cutaneous T-cell lymphomas [50]. JunB is an essential transcription factor for the differentiation of inflammatory T-helper 17 (Th17) cells [51,52,53], which demonstrates that JunB plays roles in T-cell programming. A case study on leukemia demonstrated that JunB expression levels significantly decreased in human chronic myelogenous leukemia (CML) [54]. Abnormally expressed JunB transactivated by IL-6/STAT3 signaling promotes uveal melanoma aggressiveness via epithelial–mesenchymal transition [55]. The role of JunB in psoriasis-like skin disease and arthritis was also reported. The JunB loss in keratinocytes induces chemokine/cytokine expression, attracts neutrophils and macrophages to the epidermis, and contributes to phenotypic changes in psoriasis [56]. A recent study using integrated bulk and single-cell RNA sequencing identified disease-relevant monocytes and a gene network module underlying systemic sclerosis (SSc). Four inflammatory genes from CD16+ monocytes, including JunB, showed the greatest differential expression between SSc and the healthy controls [57]. The defective degradation of JunB in patients with systemic sclerosis contributes to the overproduction of type I collagen and the development of dermatofibrosis [58].

JunB has been implicated in angiogenesis, and its expression is induced by hypoxia and VEGF. In endothelial cells, JunB regulates endothelial cell functions as a downstream factor of VEGF signaling [33]. Moreover, JunB is a hypoxia-inducible factor, its levels are elevated via the translocation and activation of NF-κB under hypoxic conditions [59]. In addition to transcriptional regulation, JunB activities are regulated via its translational regulation and phosphorylation [48,60]. VEGF is an upstream regulator of JunB [33] and also JunB regulates VEGF transcription [26,27,59]. Thus, JunB is implicated in angiogenesis by controlling the transcription upstream and downstream of VEGF-signaling. Mechanistically, the VEGF promoter contains two AP-1 transcription factor-binding sites, and induced JunB binds to the VEGF promoter to positively regulate VEGF expression under hypoxia [59]. In HUVEC, JunB induces VEGF expression, miR-3133 functions as a negative regulator of the JunB/VEGF pathway, thereby affecting angiogenesis [27]. There have also been reports of epigenetic regulation, specifically, JunB exhibits protein–protein interaction with BRG1 in the target gene promoter in HeLa cells [61,62]. Yeast two-hybrid screening has shown that BAF60a of the SWItch Sucrose non-fermentable (SWI/SNF) complex binds to c-Jun, JunB, and c-Fos [63]. In addition, it was reported that breast cancer type 1 susceptibility protein (BRCA1) interacts with JunB and regulates the transcription activity of the activation domain (AD) of BRCA1 in HEK293T cells [64]. In particular, the coiled-coil domain of BRCA1 interacts with the basic leucine zipper (bZIP) domain of JunB, and this interaction enhances BRCA1 AD activity, which affects BRCA1 transcription activity in HEK293T cells [64]. BRCA1 plays important roles in maintaining chromatin stability via its functions in transcriptional regulation and DNA repair [65,66]. BRCA1 is also known to interact with SWI/SNF chromatin remodeling complexes in breast cancer [67]. In retinal vascular development, the genome-wide accessibility of AP-1-binding sites is epigenetically controlled by sphingosine-1-phosphate (S1P) signaling. This process alters the chromatin composition of the AP-1-binding site to become closed and inaccessible, resulting in the inhibition of JunB-related gene expression during vascular maturation in the mouse retina [68].

JunB is expressed in endothelial cells and regulates their morphogenesis by regulating the core-binding factor beta subunit (CBFβ), which controls MMP13 expression, cell migration, and tube formation [42]. In this case, both the JunB–JunB homodimer and JunB–ATF2 heterodimer regulate CBFβ expression in endothelial cells [42].

### 2.2. JunB Is a Tip Cell Factor in Response to VEGF Signaling

In the sprouting region of the primordial vascular plexus, vascular endothelial cells located at the tip of the growing vessel extend numerous filopodia toward the vessel-free region. The endothelial cells that extend these filopodia are termed tip cells, and the endothelial cells that proliferate behind the tip cells and support vascular growth are called stalk cells [69]. The morphology of tip cells is similar to that of the growth cone in nerve axon elongation. Tip cells migrate by extending their filopodia in response to the concentration gradient of VEGF expressed in vascular-free areas, which determines the direction of vascular growth. However, stalk cells actively proliferate and define the number of cells in the subsequent vascular network. Tip cells and stalk cells maintain plasticity in the construction of the vascular network and can switch morphology. The tip cell and stalk cell specification mechanisms have been clarified [9]. Tip cells express the DLL4, while stalk cells express the Notch receptor Notch1. This DLL4-Notch1 signaling regulates the equilibrium between the tip and stalk cells, as well as the extension of the filopodia of tip cells to maintain a proper vessel density. Therefore, in DLL4-knockout mice, the number of filopodia extending from the tip cells increases along with the ratio of tip cells to stalk cells, resulting in the formation of hyperplasia of the vascular network with compressed vessel spacing. Nevertheless, the growth rate of blood vessels decreases. Eventually, with the emergence of blood flow, the primitive vascular plexus infiltrates the tissue and undergoes a process termed remodeling, resulting in the emergence of a hierarchical mature vascular network.

JunB is reportedly upregulated in tip cells at the angiogenic frontier and contributes to vascular development in mouse embryonic skin [26] and retinal vasculatures [68] (Figure 1). The induction of JunB expression is temporal and spatial in tip cells at the angiogenic frontier or at the branching points during vascular development. In vitro analysis using human primary microvascular endothelial cells (HMVEC) showed that the induced expression of JunB results in marked changes in cell morphology [26]. Specifically, JunB expression has a profound effect on the cell morphology of vascular endothelial cells, causing them to change to a fibroblast-like spindle cell morphology. This morphological change can be positively or negatively regulated by controlling JunB expression alone. In addition, studies using a three-dimensional collagen matrix angiogenesis assay demonstrated that JunB-expressing vascular endothelial cells enhance cell motility and regulate vascular formation. In the development of retinal blood vessels, JunB is strongly induced in the characteristic tip cells located at the angiogenic frontier of the process of radial extension of the primitive vascular plexus into retinal tissue. When JunB expression is abolished by conventional JunB knockout or endothelial cell-specific knockout in mice, the number of tip cells decreased in retinal vasculature, which resulted in a marked suppression of vascular progression and branching [68].

## 3. Endothelial JunB Functions in Vascular Development In Vivo

### 3.1. JunB Is Required for Placentation and Heart Vascular Development in Mice

The in vivo functional study of JunB using conventional JunB knockout mice was the first to describe the loss of JunB resulting in embryonic lethality through placental malformation and failure of cardiac vasculogenesis [70]. More specifically, mouse embryos with conventional JunB knockout die between E8.5 and E10 due to the abnormal formation of the placenta. Knockout of JunB does not affect cell proliferation, however, embryo growth was retarded due to a failed connection of maternal circulation. These results indicated that JunB is a transcription factor required for correct vascular development [42,59,70].

### 3.2. JunB Is Involved in Retinal Vascular Outgrowth and Retinal Special Vascular Differentiation

Retinal vasculature has been widely used as an analysis system for angiogenic sprouting and vascular development due to its postnatal development and ease of tissue accessibility. VEGF plays a central role in retinal vascular development [8], and it has been shown that VEGF receptor 2 (VEGFR2) and VEGFR receptor 3 (VEGFR3) are required for vessel sprouting [71,72]. VEGFR3 is particularly highly expressed in tip cells and induces angiogenesis and vessel growth in the absence of VEGFR2 in a Notch-independent manner [73]. The VEGFR3 signaling pathway is essential for the development of angiogenesis, with VEGFR3 expression being characteristically induced in developing vascular endothelial cells and is later restricted in lymphatic endothelial cells [74,75]. VEGFR3, which binds to VEGFC and VEGFD, is essential for angiogenesis, particularly in the developmental stages [76] (Figure 2, right). Recently, it was shown that JunB is involved in the pathological process of hypoxia-mediated retinal neovascularization [77]. In retinal endothelial cells, VEGFA signaling phosphorylates intracellular protein kinase C theta (PKCθ), and retinal endothelial cells form tip cells, thereby positively regulating cell migration, sprouting, and tube formation. In hypoxia-induced VEGFA signaling, PKCθ phosphorylation upregulates JunB expression, which induces VEGFR3 expression, thereby inducing tip cell formation, sprouting, and neovascularization of retinal capillaries via STAT3 activation. VEGFR3 is a key regulator of retinal neovascularization, while endothelial-specific knockout of JunB inhibits VEGFR3 expression, inhibiting retinal neovascularization. Mechanistically, it was shown that JunB acts downstream of PKCθ, and JunB directly binds to the promoter of VEGFR3 to regulate VEGFR3 expression (Figure 2).

Another study on retinal angiogenesis demonstrated that JunB is involved in the process of retina-specific vascular network development [68]. Retinal blood vessel networks are produced by vascular endothelial cells that invade into the eye retina tissue from the optic nerve papilla. In the early stage of retinal vascular development, radial retinal structures are produced along the superficial nerve fiber layer of the retina to form the superficial plexus. Later, around P8 in mice, retinal blood vessels begin to invade into the retinal nerve tissue vertically toward the retinal deep layer, forming a deep plexus, which is followed by the formation of the middle layer plexus. Yanagida et al. focused on S1P receptor signaling in retinal vascular development and found that it regulates the open-chromatin state of the AP-1 binding motif of retinal endothelial cells genome-wide [68]. Postnatally, following the complete abolishment of S1P receptor signaling by the receptor triple knockout, the AP-1 motif in the open-chromatin region was strongly accumulated. Furthermore, the investigators confirmed that JunB was the AP-1 transcription factor involved in this process. JunB was highly expressed in tip cells at the tip of angiogenesis and promoted angiogenesis until the activation of S1P receptor signaling. Following blood perfusion, the serum-derived S1P induces vascular endothelial-cadherin (VE-cadherin) expression in endothelial cells and results in the suppression of JunB expression in endothelial cells. At the same time, S1P also induces chromatin remodeling to close the AP-1 motif in endothelial cells, which have a lumen structure with blood flow. Reportedly, VE-cadherin is induced by S1P signaling in mature retinal vasculature with lumen structures where blood flow is present, resulting in decreased expression of JunB. In other words, S1P signaling converts the tip cell-like state activated by JunB to a retinal-specific vascular maturation state. Interestingly, in retinal angiogenesis, JunB expression is activated at the sprouting point, i.e., at the branching point where blood vessels extend vertically into the deep retinal layers. Although the initiation of vertical branching is unclear, some stimuli (partial weakening of S1P signaling or interaction of endothelial cells with retinal neurons) causes JunB induction and chromatin remodeling in vascular endothelial cells, thereby exposing the AP-1 site. As a result, JunB target genes (including tip cell genes) can be induced, and the cell converts to a tip cell, extending the retinal plexus invasion into the deeper layers of retinal tissue. During developmental vasculogenesis, JunB expression is specifically observed in endothelial cells. The role of JunB in this directional angiogenesis during retinal vasculature development is consistent with its function in cutaneous vascular development, as described below.

### 3.3. JunB Regulates Neurovascular Parallel Alignment in Developing Skin Vasculture in Mice

It is established that nerves and blood vessels align parallel to each other. However, until recently, the molecular mechanism involved in this process was unclear. Recent reports described that both neuronal signaling to the blood vessels [78] and vascular signaling to the nerves [79] are involved in the regulation of this juxtaposition process. Moreover, the nerves and blood vessels co-operatively form mature and correct neural and vascular network structures. In zebrafish, it has been reported that inhibition of the function of VEGFR3 results in the loss of nerve-vessel parallelism. Furthermore, the trapping of VEGFC secreted by arterial vascular endothelial cells using VEGFR3-Fc recombinant protein results in abnormal aortic ventral motor nerve function [80]. These findings suggested that VEGFC-VEGFR3 signaling plays an important role in nerve-vessel parallelism. In the development of the subcutaneous vascular network in mouse embryos, peripheral nerve fiber bundles align parallel to arterial blood vessels while retaining a certain distance from nerves, and form a mature vascular network. In 2013, Li et al. reported that VEGF and CXCL12 secreted by nerves act on the primitive vascular plexus, which is the initial structure of blood vessels. This induces vascular remodeling to form neurovascular parallelism, resulting in the formation of a proper mature vascular network [81]. Using a co-culture system of dorsal root ganglion cells and primary HMVEC, it was found that JunB was strongly and specifically induced in endothelial cells during neurovascular interactions [26]. In addition, the knockdown of JunB resulted in disruption of the neurovascular parallelism in vivo. This evidence indicated that induction of JunB in endothelial cells is required for neurovascular parallel alignment during vascular development in mouse embryonic skin [26]. Furthermore, we found that JunB induction is mediated by direct cell-cell contact with nerves-vessels rather than soluble factors secreted by the nerves. Interestingly, vascular endothelial cells with force-induction of JunB expression adopt a spindle-like morphology, show increased motility, and exhibit a unique phenotype which is very similar to that of tip cells found at the angiogenic frontier. In fact, vasculogenesis assays using a three-dimensional collagen matrix and endothelial sphere outgrowth assays revealed that JunB expression is activated in tip cells at the angiogenic frontier [26]. It is clear that JunB is involved in vascular remodeling by conversion of vascular endothelial cells to tip cells, and JunB expression is required for the juxtapositional alignment of the blood vessel to neurons (Figure 2, left). However, the mechanisms through which directional remodeling is induced remains unclear. The candidate molecule(s) of this mechanism may be the upstream molecule(s) of JunB, which are supplied by nerves at the neurovascular interaction.

### 3.4. JunB also Controls Lymphangiogenesis

During lymphatic vessel development in vertebrates, Prospero homeobox protein 1 (PROX1) expressing endothelial cells in cardinal veins eventually differentiate into lymphatic endothelial cells during lymphangiogenesis, which is considered the master regulator of lymphatic endothelial cell fate specification [82,83]. Forkhead box protein O1 (FOXO1) is required for proper lymphatic vessel development and maturation by upregulating C-X-C chemokine receptor 4 (CXCR4) expression in lymphatic endothelial cells in mouse tail dermis [84]. The functions of JunB in lymphatic vascular development in zebrafish have been described by Kiesow et al. [85]. JunB directly regulates miR-182 expression, which downregulates its downstream FOXO1 expression in lymphatic endothelial cells. The loss of JunB leads to the failure of parachordal lymphangioblast and thoracic duct formation in zebrafish, indicating that JunB plays an important role in lymphatic vascular morphogenesis in zebrafish by negatively regulating JunB/miR-182/Foxo1 axis signaling. The endothelial-specific deletion of Foxo1 in mice results in embryonic lethality at approximately E10.5 due to vascular remodeling defects [86,87]. Its phenotype is similar to that of JunB knockout mice [70]. Nevertheless, currently, the involvement of the JunB–miR-182–FOXO1 axis in blood vessel development remains unknown. In addition, VEGFC–VEGFR3 signaling is implicated in the survival, proliferation, and migration of lymphatic endothelial cells [88,89]. Thus far, the role of JunB in the regulation of VEGFR3 expression in lymphatic endothelial cells is unclear.

## 4. Conclusions

The main original articles that describe the functions of JunB in angiogenesis and vascular development, discussed in this review, are summarized in Table 1. A variety of angiogenesis-related factors have been identified, including exogenous angiogenesis-inducers (e.g., hypoxia and VEGF), transcription factors in vascular endothelial cells that are induced in response to these inducers, and angiogenesis-related genes transcribed by these transcription factors. Some transcription factors induce the expression of tissue-specific endothelial cell phenotypes in specific endothelial cells. In angiogenesis, tip cells appear at the tip of angiogenesis after receiving angiogenesis-inducing cues (e.g., VEGF) and induce sprouting with migration of vascular endothelial cells to form a proper vascular network. The expression of AP-1 transcription factor JunB is induced in tip cells during vascular development of skin capillaries and retinal vessels, leading to the induction of tip cell-specific characteristics (e.g., cell migration and angiogenic sprouting).

The mechanism of vascular “directional” control appears to be maintained mainly through the signaling balances between attractive and repulsive effectors from various tissues during vascular formation. For example, arterial and venous juxtapositional alignment is controlled by the balance of two types of action: Repulsive effects between arterial EphrinB2 and venous EphB4 [90,91] and attractive action between arterial apelin and the venous apelin receptor (APJ) [92]. The transient induction of tip cells at the frontier of angiogenesis has a dynamic response to various angiogenic attractors (e.g., hypoxia and VEGF), which elongate and remodel the blood vessels according to the location and timing to form a proper vascular network. We and others have shown that JunB induction in tip cells is a novel angiogenic machinery that regulates the motility, filopodia formation, invasion, and remodeling of tip cells in response to angiogenic cues. By controlling the temporal and spatial induction of JunB in tip cells, we may be able to control the elongation of blood vessels. This suggests that the tip cell factor JunB is a determinant of vascular directionality. Nevertheless, the involvement of JunB in tip-stalk cell sorting signaling by Notch-DLL4 intercellular signaling remains unclear.

In the process of retinal vascular development, JunB is specifically upregulated in tip cells at the angiogenic frontier in the upper layer of retinal nerve tissue and migrates radially on the upper layers of the retina. Expression of JunB is also observed specifically in tip cells vertically protruding into the deep retinal plexus layer, leading to the formation of longitudinally oriented retinal blood vessels [68]. In the remodeling process of mouse embryonic skin vascular development, JunB is induced in endothelial cells in contact with the nerves and enhances the invasiveness of the endothelium to create a vascular network that aligns parallel to the nerves [26]. A common feature of JunB-mediated vascular network formation in both embryonic skin and retina vasculature is that the regulation of the directional vascularization is based on the interaction with nerve cells. Probably, the angiogenic attractors of extracellular matrix produced by neurons or neuronal membrane proteins play a role in determining the direction of blood vessels. In response to this, JunB is induced in vascular endothelial cells to increase migratory activity and determine the direction of extension. However, at present, the factor(s) on the neural side that play a role in this guidance are unknown. Hence, further extensive research is warranted.

## Figures and Tables

**Figure 1 ijms-22-02804-f001:**
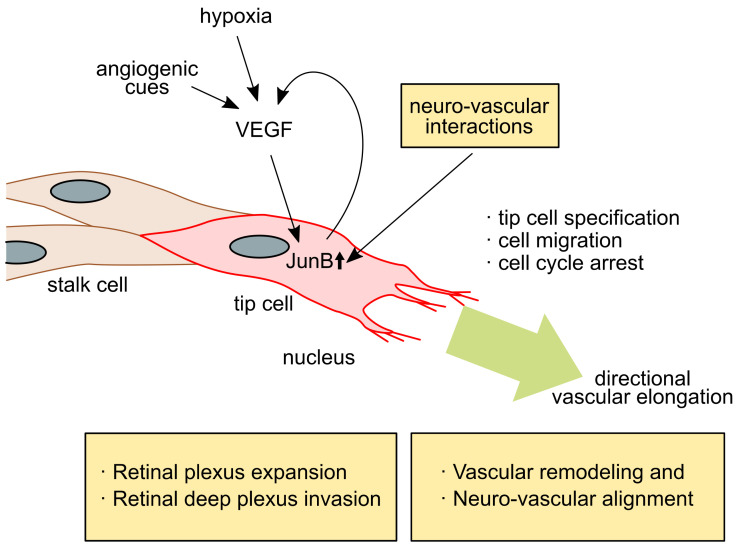
In the event of angiogenesis in embryonic skin and retina, endothelial cells respond to angiogenic cues, such as hypoxia or other signals, to produce new vessel branches at the angiogenic frontier. The vascular endothelial growth factor (VEGF) plays a central role in this process. The VEGF signals that induce JunB expression (small upper arrow) result in the conversion of endothelial cells to tip cells. JunB activation is involved in the vessel parallel alignment with neurons in developing skin, retinal tissue-specific radial vessel expansion, and deep plexus expansion in retinal vessel development. Both vessel-wiring processes in embryonic skin and the retina include neurovascular interactions. The arrows indicate the relationship of signaling directions. Large green arrow indicates direction of vascular elongation.

**Figure 2 ijms-22-02804-f002:**
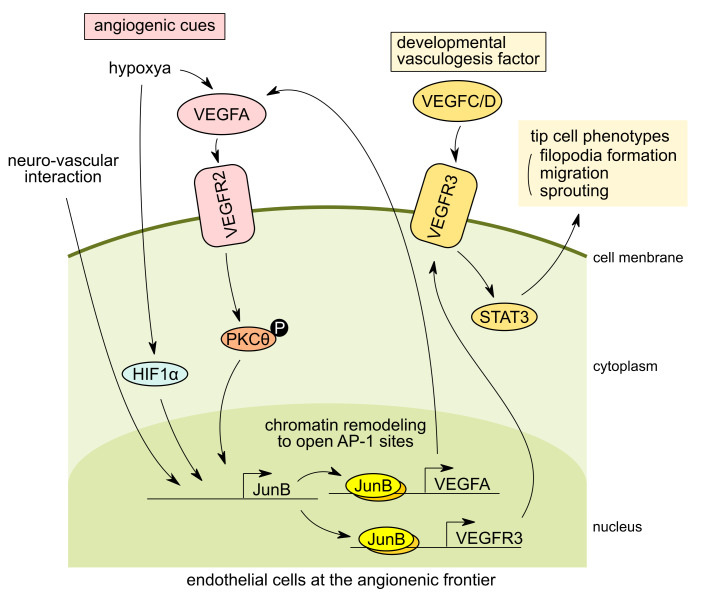
Tip cell phenotype expression in response to angiogenic cues in endothelial cells at the vasculogenesis frontier. In endothelial cells at the angiogenesis frontier during vascular development, JunB expression is induced by angiogenic cues including hypoxia and vascular endothelial growth factor A (VEGFA) through hypoxia-inducible factor 1 subunit alpha (HIF1α) or phosphorylation of protein kinase C theta (PKCθ). “P” indicates phosphorylation of the molecule. (left). In the developing skin vasculatures, the direct interaction of endothelial cells with peripheral neurons can also stimulate JunB expression and coordinate neurovascular parallel alignment (left). In tip cells, JunB is involved in the upregulation of vascular endothelial growth factor receptor 3 (VEGFR3) expression which is a key VEGFR for angiogenesis at specific developmental stages. VEGFR3 binds vascular endothelial growth factor C and D (VEGFC/D), while signaling cascades are required for tip cell migration and sprouting of endothelial cells by signal transducer and activator of transcription 3 (STAT3) activation at the angiogenic frontier (right). The arrows indicate the relationship of signaling directions.

**Table 1 ijms-22-02804-t001:** Original articles which describe JunB functions in angiogenesis and vascular development.

JunB Functions in Endothelial Cells	Reference
Angiogenesis	[26,27,42,59,68,70,77]
Neurovascular parallel alignment	[26]
Filopodia formation and tip cell specification	[26,68,77]
Retinal vascular development	[68,77]
Lymphangiogenesis	[85]

## Data Availability

Not applicable.

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
