# Peer review of "Emerging Role of AP-1 Transcription Factor JunB in Angiogenesis and Vascular Development"

_ijms, 2021, doi:10.3390/ijms22062804_

Round 1

Reviewer 1 Report

The authors should consider the followings, whilst they aimed in a comprehensive review describing the role JunB in angiogenesis and vascular development.

  1. The authors may expand their review in the lymph area.
  2. Other than human cell lines, the authors may need to review if JunB had any indicated value in human tissue/ organ levels, of their developmental, normal and/or pathological settings.
  3. BRCA1 seems to be associated with JunB, why the authors not included this linkage in the review?
  4. The authors may also summarize their review findings in a table form.
  5. Upon Figure 1 and Figure 2, the authors may consider the color option for a better illustration.
  6. The authors shall increase the number of articles referred and to cite more recent works in the field.
  7. In the Figure 2, the authors may outline the cellular structure, such as indication of cell membrane, nucleus, in the drawing.
  8. The authors may consider using English proof-reading services by language professional.

Author Response

Thank you for your comments and suggestions regarding our manuscript. We are grateful for your detailed review. Newly added or revised text has been indicated in yellow highlights in the revised manuscript. Our responses are provided below.

[The authors should consider the followings, whilst they aimed in a comprehensive review describing the role JunB in angiogenesis and vascular development.]

[1. The authors may expand their review in the lymph area.]

Thank you for your suggestion. We have expanded our description of lymphangiogenesis and JunB in section 3.4, lines 333–339 and 343–347.

[2. Other than human cell lines, the authors may need to review if JunB had any indicated value in human tissue/ organ levels, of their developmental, normal and/or pathological settings.]

We have included descriptions and references for human JunB expression levels in several normal and pathological conditions (lines 121–140).

[3. BRCA1 seems to be associated with JunB, why the authors not included this linkage in the review?]

Thank you for pointing out that the description of BRCA1 was missing. We have included a description regarding the regulation of BRCA1 activities by binding with JunB (lines 156–164).

[4. The authors may also summarize their review findings in a table form.]

We have added a new Table 1 that summarizes articles on JunB function in angiogenesis (lines 366–367).

[5. Upon Figure 1 and Figure 2, the authors may consider the color option for a better illustration.]

Thank you for your suggestion. We have presented Figures 1 and 2 in color for better illustration.

[6. The authors shall increase the number of articles referred and to cite more recent works in the field.]

By increasing description on human JunB levels on page 3, the number of referenced articles increased from 73 to 92. The references include 8 latest articles published in 2020–2021.

[7. In the Figure 2, the authors may outline the cellular structure, such as indication of cell membrane, nucleus, in the drawing.]

Thank you for your suggestion. We have fixed Figure 2.

[8. The authors may consider using English proof-reading services by language professional.]

Thank you for your suggestion. We have rechecked and corrected several sentences throughout the manuscript. In addition, the manuscript has been carefully reviewed by an experienced professional whose first language is English and who specializes in editing scientific papers. We hope the descriptions throughout the manuscript are suitable for publication in International Journal of Molecular Sciences.

Reviewer 2 Report

The manuscript titled “ Emerging Role of AP-1….vascular Development” authored by Yoshitomi et al. reviews the literature published so far on the role of AP-1 transcription factor JunB in angiogenesis and vascular development. The authors systematically highlight the emerging role of JunB in tissue-specific vascular maturation processes in neuro-vascular interactions in murine skin and retinal vasculatures. The review is concise, well written and provides an overview of the role of the AP-I transcription factor JunB in endothelial cells during vascular development. In its present form, the manuscript is acceptable for publication.

Author Response

Thank you for your comment on our manuscript. We are grateful for your detailed review of our manuscript. We hope that our revisions will meet your approval. Please let us know if any additional information or adjustments are needed.